

# Assessment of optimal empty flushing strategies in a multi-reservoir system

## Frederick N.-F. Chou[1] and Chia-Wen Wu[1]

[1] Department of Hydraulic and Ocean Engineering, National Cheng-Kung University, 1 University Rd., Tainan, Taiwan

Correspondence to: Chia-Wen Wu (chiawenwu1977@gmail.com)

**Abstract**

Empty flushing is the most effective approach to evacuate the deposited sediments in

the reservoir. However, emptying reservoir essentially conflicts with its water supply operation, thus a feasible strategy of empty flushing should prevent significant increase of water shortage risks. This paper presents a framework of performing empty flushing in a multi-reservoir system, where flushing is carried out in a primary reservoir and the other reservoirs provide backup storage for stable water supply during flushing. A network flow

programming-based model is employed to simulate daily joint operation of reservoirs. During the simulation, if the storage of each reservoir achieves the predefined conditions, drawdown and empty flushing of the primary reservoir is activated. During the flushing, if the storage of any reservoir reaches the pre-defined thresholds, then the flushing operation is halted and the simulation switches back to the regular joint operation mode. This simulation

model is linked with a nonlinear optimization algorithm to calibrate the optimal parameters. The optimized strategy yields a maximum amount of flushed sediments, while the incremental water shortage is controlled within the acceptable threshold.

**Keywords***: reservoir desilting, empty flushing, water supply, joint operation of multiple reservoirs





## 1. Introduction

Empty flushing is the most effective method for removing deposited sediments from reservoirs (Fan and Morris, 1992; Morris and Fan, 1998; Shen, 1999). This process requires complete drawdown of reservoir storage to allow "inflows to pass through at riverine depths" (Atkinson, 1996). The drawdown of storage is usually carried out by releasing water through bottom outlets, such as sluiceways. During this process, the accelerated flow near the inlet may partially reactivate and scour out the depositions to generate a flushing cone in the vicinity of the inlet. By completely emptying the reservoir and maintaining the riverine flow condition, retrogressive erosion may be induced from the rim of the flushing cone extending to the upstream to create a flushing channel. The formation of the flushing channel usually leads to hyper sediment concentration of the bottom release and thus effectively recovers partial deposited capacity of the reservoir. This operation has been used to pursue sustainable utilization by many reservoirs worldwide (Atkinson, 1996; White, 2001; Chaudhry and Habib-ur-rehman, 2012), some examples of which are presented in Table 1.

The other side to desilting, however, is that draining the storage of a reservoir counteracts its water supply function. Hence empty flushing is generally limited to reservoirs that operate solely for hydropower generation, flood mitigation, or irrigation. These purposes usually do not require reservoir storage during certain periods of the year, during which empty flushing can be implemented without impairing the original design function of the reservoir. However, for reservoirs with municipal or industrial water users that rely on sufficient storage for steady water supply, the implementation of empty flushing is relatively rare.

The conflict between water supply and empty flushing has been addressed by Chang et al. (2003) and Khan and Tingsanchali (2009). Chang et al. (2003) developed the operating



rule curves and empty flushing schedule for the Dapu Reservoir in central Taiwan. A genetic algorithm was used to optimize the rule curves by minimizing the shortage index for irrigation with consuming water for empty flushing only during prescribed periods. Khan and Tingsanchali (2009) applied a similar approach to the Tarbela Reservoir in Iran. The objective function in deriving rule curves is to maximize the net benefit from irrigational water supply, hydropower generation, sedimentation evacuation and flood mitigation. However, these previous studies dealt only with single-reservoir. If there are additional reservoirs in the system that can act as backup water sources, it may be possible to elevate the feasibility of empty flushing by reducing its impact on water supply through appropriate joint operation. The ideal operating strategy may require utilizing the reservoir with the most excessive sediment deposition, referred to as the primary reservoir throughout the remainder of this paper, to supply demands while preserving the storage in the other reservoirs before empty flushing. This will lead to a lower water surface level (WSL) in the primary reservoir and a higher WSL in the others. Empty flushing in the primary reservoir can then be activated once favorable conditions are achieved, such as adequate storage distribution among reservoirs to ensure both high sediment flushing efficiency and steady backup water supply.

This study focused on a water resources system that contains multiple reservoirs, among which one primary reservoir requires appropriate empty flushing operation while the others could provide backup storage. The goal is to develop the optimal strategy, which maximizes the efficiency of empty flushing, for such a system. In the following section, key factors influencing the efficiency of sediment flushing as well as the stability of water supply are discussed. The methodology to derive the optimal strategy for joint water supply operation and empty flushing in a multi-reservoir system is then presented. The proposed





approach is then applied to a tandem online and offline reservoir system in southern Taiwan.

The results from the case study validate the efficacy of the derived optimal strategy.





## Table 1 Cases of empty flushing

| Reservoir | Country | Effective capacity (Mm3) | Major Purpose | Number of flushing days per operation | Reservoir system | Watershed sediment yield (M.ton/year) | Capacity-inflow ratio (CIR) | Annual flushing periods | Backup water supply during flushing | Flushing facilities | Flushing experiences | Literature source |
|---|---|---|---|---|---|---|---|---|---|---|---|---|
| Akung-ten | Taiwan | 16.47 | FC IR | 100 days | S | 0.55 | 0.71 | June to September | Trans-basin diversion from an adjacent river | morning glory with capacity of 85 m³/s | Flushing out 5% to 54% of the inflowing sediments during floods between 2009 to 2013 | Southern Water Resources Office, 2013 |
| Baira | India | 2.40 | HP | 1 to 2 days | S | 0.3 (from siltation after 18 months) | 0.001 | April to May | Halting hydro-power generation | low–level diversion tunnel with capacity of 44 m3/s | The first operation in Aug of 1983, lasting for 40 hours with discharge of 44 m3/s, flushed out 85% of the deposited sediments. Afterwards the empty flushing is suggested to be annually performed during April to May. | Jaggi and Kashyap, 1984; Atkinson, 1996; Chaudhry and Habib-ur-rehman, 2012 |
| Cachi | Costa Rica | 54.0 | HP | 2 to 3 days | S | 0.81 | 0.016 | May (the beginning of wet season) | Halting hydro-power generation | Bottom outlet | On average flushing out 0.25 million m3/year of sediments annually. | Brandt and Swenning, 1999; Jansson and Erlingsson, 2000; Chaudhry and Habib-ur-rehman, 2012; |
| Dapu | Taiwan | 5.29 | IR ID | 50 days | S | 0.40 | 0.04 | May to July | Halting irrigational supply. Industrial demand is supplied by reservoir inflow | Sluiceway with capacity of 325 m3/s | On average flushing out 0.20 million m3/year of sediments annually. | Chang et al, 2003; Water Resources Agency, 2010 |
| Gebidem | Switzerland | 9.0 | HP | 2 to 4 days | S | 0.50 | 0.02 | May to June | Halting hydro-power generation | Bottom outlet with flushing discharge of 10 m3/s | Since 1992, the annual volume of flushed sediments is between 0.2 to 0.5 million m3 per year. | Atkinson, 1996; Chaudhry and Habib-ur-rehman, 2012; Meile et al. 2014 |

HP: hydropower generation, FC: flood control, IR: irrigation, ID: industrial water supply, S: single reservoir system, M: multi-reservoir system,

Capacity-inflow ratio: the ratio between the effective capacity and the annual inflow volume of the reservoir,




## Table 1 Cases of empty flushing (continued)

| Reservoir | Country | Effective capacity (Mm3) | Major Purpose | Number of flushing days per operation | Reservoir system | Watershed sediment yield (M.ton/year) | Capacity-inflow ratio (CIR) | Annual flushing periods | Backup water supply during flushing | Flushing facilities | Flushing experiences | Literature source |
|---|---|---|---|---|---|---|---|---|---|---|---|---|
| Hengshan | China | 13.30 | FC IR | 10 to 20 days | S | 1.18 | 0.84 | June to September | Diverting turbid release to provide irrigational demand. | Bottom outlet with capacity of 17 $m^3$/s at full impounding level and 2 $m^3$/s during empty flushing | The first operation in 1974 lasted for 37 days and flushed out 0.8 million $m^3$ of sediments. The second operation in 1979 lasted for 52 days and flushed out 1.03 million $m^3$ of sediments. | Atkinson, 1996; Chaudhry and Habib-ur-rehman, 2012 |
| Jensan-pei | Taiwan | 1.51 | IR | 53 days | S | 0.28 | 0.80 | May to June | Halting water supply | Sluiceway with capacity of 12.2 $m^3$/s | On average flushing out 0.33 million $m^3$/year of sediments annually. | Water Resources Planning Institute, 2010 |
| Manga-hao | New Zealand | 2.39 | HP | 30 days | M | — | — | May | Halting hydro-power generation | low-level diversion tunnel | During the total duration of one month of flushing in 1969, 0.8 million $m^3$ of sediment has flushed from the reservoir, which equals to the 75% of sediment that had accumulated since 1924 | Jowett, 1984; Atkinson, 1996; White, 2001; Chaudhry and Habib-ur-rehman, 2012 |
| Nan-qin | China | 10.20 | IR FC | 4 days every 3-4 years | S | 0.53 | 0.08 | The end of the flood season | --- | Sluiceway with flushing discharge of 14 $m^3$/s | The first operation in 1984 flushed out all inflow sediments in 1984, along with 0.72 million m3 that had deposited in the earlier years. | Chen and Zhao, 1992; Chaudhry and Habib-ur-rehman, 2012 |
| Santo Domingo | Venezuela | 3.00 | HP | 3 to 4 days | S | 0.20 | 0.01 | May | Halting hydro-power generation | Three bottom outlets with capacity of 13 $m^3$/s | The first operation in May of 1978 lasted for 4 days and flushed out 50-60% of the deposited sediments. Afterwards mechanical excavation was used to disperse the consolidated deposits and empty flushing is again performed for three weeks to fully restore the deposited capacity. | Krumdieck and Chamot, 1979; Atkinson, 1996 |

HP: hydropower generation, FC: flood control, IR: irrigation, ID: industrial water supply, S: single reservoir system, M: multi-reservoir system,

Capacity-inflow ratio: the ratio between the effective capacity and the annual inflow volume of the reservoir,



**Table 1 Cases of empty flushing (continued)**

| Reservoir | Country | Effective capacity (Mm3) | Major Purpose | Number of flushing days per operation | Reservoir system | Watershed sediment yield (M.ton/year) | Capacity-inflow ratio (CIR) | Annual flushing periods | Backup water supply during flushing | Flushing facilities | Flushing experiences | Literature source |
|---|---|---|---|---|---|---|---|---|---|---|---|---|
| Sefid-Rud | Iran | 1760 | IR HP | 4 months | S | 50 | 0.36 | October to February | No requirement for irrigational water supply | Bottom outlets with flushing discharge of 100 $m^3$/s | Empty flushing during non-irrigational periods removes approximately 28.4 million T of sediments per year. | Atkinson, 1996; Taklimy and Tolouie, 2005 |
| Zemo-Afchar | Former USSR | -- | HP | 1 to 3 days | S | Suspended load 4 Mm$^3$ | -- | April, May or November | Halting hydro-power generation | Bottom outlets with flushing discharge of 450 $m^3$/s | Implemented from 1939, with full drawdown. Removing about 1.0 million $m^3$ (from 0.5 to 2 million $m^3$) per year | Bruk, 1985; Chaudhry and Habib-ur-rehman, 2012; |
| Dashidaira | Japan | 1.657 | HP FC | 1 to 2 days | M | 0.62 | 0.00674 | June to August | Halting hydro-power generation | Bottom outlets with flushing discharge between 200~300 $m^3$/s | When inflow at the upstream Dashidaira Dam exceeds 300 $m^3$/s at the first time of the year during June to August, a coordinate flushing is performed. The average annual flushed volume between 2001 to 2007 is 0.27 million $m^3$/year | Sumi, 2008; Sumi et al., 2009 |
| Unazuki | | 12.70 | | | | 0.96 | 0.014 | | | | | |
| Verbois | Switzerland | 12.00 | HP | 1 to 2 days every 3 years | M | 0.33 | 0.00144 | May to June | Halting hydro-power generation | Bottom outlets with flushing discharge of 600 $m^3$/s | Flushing is performed in every 3 years. The volumetric flushed sediments per event is around 0.6 and 1.1 million $m^3$ for Verbois Reservoir and 0.1 and 0.4 million m3 for Genissiat Reservoir according to Sumi (2008) | Sumi, 2008 |
| Genissiat | France | 18.00 | | | | 0.73 | 0.00467 | | | | | |

HP: hydropower generation, FC: flood control, IR: irrigation, ID: industrial water supply, S: single reservoir system, M: multi-reservoir system,

Capacity-inflow ratio: the ratio between the effective capacity and the annual inflow volume of the reservoir,



## 2. Material and Methods

### 2.1 Qualitative analysis: key factors for successful operations of empty flushing

Two performance indices, expected desilting volume and the induced increments of water shortage, are used in this study to evaluate an empty flushing strategy. An optimal strategy should maximize the desilting volume while maintaining the incremental shortage under an acceptable threshold. According to the cases in Table 1, key hydrological and operational factors for succeeding in these indices are described as follows:

1. Qualitative conditions for water supply (QCWS)

    (1) QCWS 1: Adequate water supply during empty flushing

    In order to satisfy this condition, episodes between periods with heavy water supply pressure, such as non-irrigation periods or when hydropower demands is low, can be utilized as windows of opportunity to implement empty flushing. One example is the Dapu Reservoir in central Taiwan, which functions primarily to serve agricultural and industrial water supply. Empty flushing of this reservoir is scheduled from May to July when irrigation demand is low and reservoir inflow is sufficient for demands. In contrast, reservoirs that provide water to the general public must maintain at stable supply level throughout the year. Empty flushing of such reservoirs would require backup water sources capable of ensuring a steady supply until the reservoir can resume normal operations. One example is the Agongdian Reservoir in southern Taiwan which undergoes empty flushing from June to September annually, during which trans-basin diversion from an adjacent basin is performed to supplement public and agricultural water supply.

    (2) QCWS 2: Adequate water supply after empty flushing





Satisfaction of this condition requires sufficient reservoir inflow following empty flushing to rapidly replenish the storage of the reservoir. Thus, if the capacity of a reservoir undergoing empty flushing is relatively small compared to the volume of its inflow can satisfy the QCWS 2. Basson and Rooseboom (1997) indicated that empty flushing is more feasible for reservoirs with an effective capacity to annual inflow volume ratio (capacity-inflow ratio, CIR) of less than 0.03. Many of the reservoirs in Table 1 fulfill this criterion. The others that have a CIR greater than 0.03 are located in areas with uneven seasonal rainfall distributions, such that the abundance of inflow during flood seasons can effectively refill the storage soon after empty flushing. One example is the Dapu Reservoir, which receives abundant rainfall and inflow from May to August every year. This particular reservoir can remain empty until early July without affecting the subsequent water supply.

2. Qualitative conditions for flushing sediments (QCFS): Compliance with these conditions promotes efficiency of sediment flushing. The key is to identify and take advantage of opportunities with both high inflow and low WSL of the reservoir by performing empty flushing.

(1) QCFS1: High inflow during empty flushing

High inflow is required to maximize the flushing efficiency by more effectively scouring the depositions of the reservoir. Atkinson (1996) and White (2001) indicated that empty flushing should only be initiated when the inflow is at least double the inflow in normal conditions. The experience with flushing the Zemo-Afchar Reservoir of the former USSR (Chaudhry and Habib-ur-rehman, 2012) suggests that empty flushing is most effective with inflow between 400 to 500 m$^3$/s, which is 2 to 2.5 times the average inflow (Bruk, 1985; Singh and McConkey-Broeren, 1990). Long-term observation of Jianshanpi Reservoir in southern Taiwan also revealed that the



efficiency of empty flushing peaks during heavy rainfall events when daily rainfall on the reservoir watershed is between 40 to 60 mm. This condition can also be artificially achieved. For instance, during the empty flushing of the Mangahao Reservoir in New Zealand, water was released from another upstream reservoir to enhance the scouring of depositions and thus maximize desilting volume (White, 2001).

(2) QCFS2: Low WSL before and during empty flushing

    a. Before empty flushing is started: During the regular operation, operators could take advantage of periods when the reservoir WSL is low to perform drawdown and initiate empty flushing. In cases where the reservoir has outlets with sufficient capacities, timely drawdowns can be performed shortly prior to expected floods so that the flood inflow can effectively scour and flush out depositions. One example is the Dapu Reservoir, of which WSL is generally the lowest in mid-May. This timing is thus considered as the ideal time to empty the reservoir, with the expectation that subsequent abundant floodwater from May to August can also fulfill the QCWS 1, QCWS 2 and QCFS 1.

    b. After empty flushing is initiated: Once empty flushing is initiated, the reservoir should remain as close to empty as possible to maintain high flushing efficiency. However, if the inflow exceeds the capacities of the outlet works, then the WSL in the reservoir will begin to rise. This leads to decreased flow velocity in the reservoir, which reduces the empty flushing efficiency. Atkinson (1996) suggested the use of the drawdown ratio (DDR) to measure the flushing efficiency. This index is defined as 1 minus the ratio between the depth of WSL during empty flushing and the depth of normal pool level of the reservoir. Atkinson (1996) and White (2001) defined incomplete drawdown flushing as situations in which DDR is less than 0.66, wherein the depth of the water during flushing is greater than a third of





the maximum depth. In such circumstances, the efficiency of empty flushing is significantly reduced and it is recommended to switch the operation to the regular mode of water supply.

### 2. 2 Quantitative derivation of the optimal empty flushing strategy

As stated in the introduction section, this study focuses on implementing empty flushing of a single primary reservoir within a multi-reservoir system. While the comprehensive discussion of the previous subsection is generally applicable to multi-reservoir systems, the proposed quantitative methodology as well as the following case study apply specifically to those systems without means to artificially generate flushing inflow to the primary reservoir. In addition, we focus on event-based operation. This means that the timing and duration of empty flushing is flexible according to real-time hydrological and operational conditions. If these conditions are not favorable, the primary reservoir could resume regular operation. This feature distinguishes the present method from previous related studies (Chang et al., 2003; Khan and Tingsanchali, 2009), which mandatorily empty reservoir storage during predetermined periods within a year. This paper also assumed that the water demands in the system require constant supply, thus rendering empty flushing infeasible during parts of the year. To facilitate determination of feasible periods for empty flushing, the following criteria are provided:

1. Meeting QCWS 1 requires periods of low water demand during which backup reservoirs can provide adequate supply during empty flushing.

2. QCFS 2 dictates that the most opportune time to begin empty flushing is at the end of the dry season. At this time the storage levels of reservoirs are usually at their lowest levels of the year. This ensures that storage can be effectively and efficiently drained by drawdown flushing through the capacitated bottom outlets of the primary reservoir.

3. Meeting QCFS 1 requires that the feasible duration for empty flushing should be extended



into the wet season to ensure adequate inflow for scouring depositions.

4. Meeting QCWS 2 requires that the feasible duration for empty flushing should be ended before the end of the wet season to ensure adequate replenishment of reservoir storage after the flushing operation.

The following proposed method for deriving optimal strategy adopts the simulation-optimization linkage approach. It requires a model to simulate the operations of water supply and empty flushing, thus allowing for quantifying the desilting volume as well as the incremental water shortage generated by a given strategy. The model simulates the process of water supply according to a set of joint operating rules as presented in subsection 2.2.1. When specific quantitative conditions presented in subsection 2.2.2 are achieved, empty flushing in the primary reservoir is activated and the approach in subsection 2.2.3 is employed to estimate the desilting volume. The empty flushing terminates when the conditions presented in subsection 2.2.4 are reached, and the simulation is switched to regular water supply operation until the next time activation conditions are satisfied. The simulation model is linked to an optimization algorithm to calibrate optimal parameters in the activation and termination conditions, according to the formulation presented in subsection 2.2.5. Fig. 1 depicts a flowchart of the analyzing procedure.





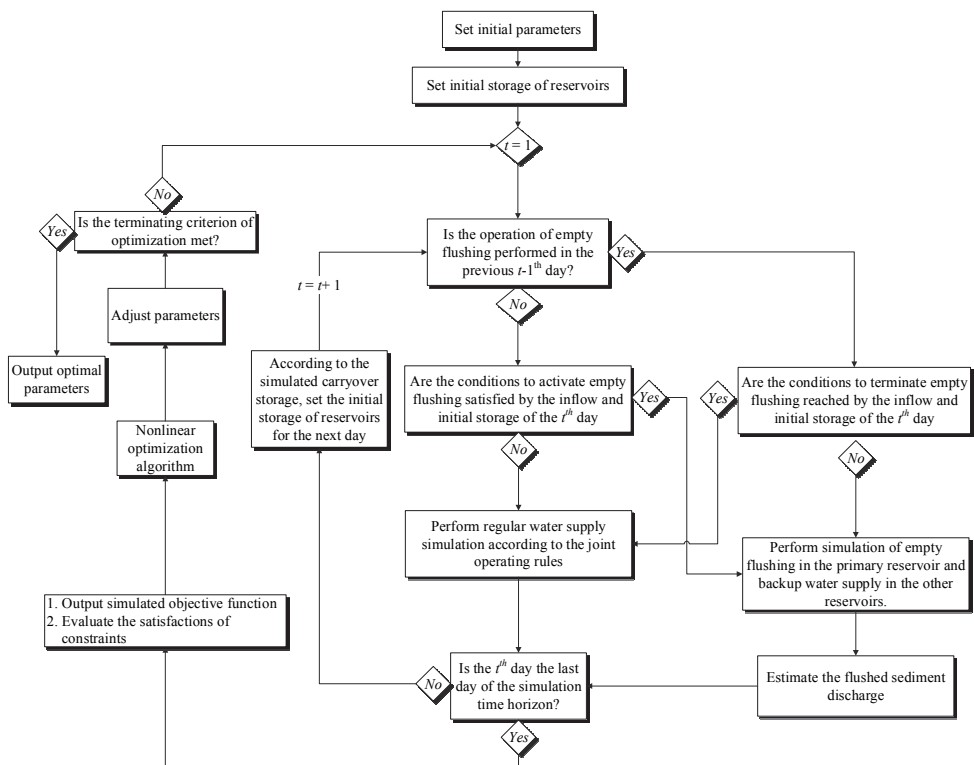

**Fig. 1 The procedure to derive the optimal empty flushing strategy**

### 2.2.1 Joint operating rules for a multi-reservoir system

5      According to Oliveria and Loucks (1997), the rules to jointly operate multiple

reservoirs for water supply include the following two phases:

1. Determination of total water supply amount: The total amount of water supply is

    determined based on the total storage of reservoirs in the system. If the total storage does

    not suffice, a discount of total water supply may be applied by the system-wide release

10   rule. Fig. 2 presents the joint operating rule curves, a form of the system-wide release rule,

    for the Tsengwen and Wushanto Reservoirs in southern Taiwan. The location and

    associated water resources system of these reservoirs are depicted in Figs. 4 and 6 in the

    case study section. The release rules stipulate that when the total storage of the two



reservoirs is below the critical limit, only 80% of the public demand and 50% of the agricultural and industrial demands will be satisfied. When the total storage is between the lower and critical limits, the public demand should be fulfilled and 75% of the agricultural and industrial demands need to be satisfied. When the total storage is between the upper and lower limits, all demands should be fulfilled. In the event that the storage in the Tsengwen Reservoir exceeds the upper limit, extra water can provide excess supply or full loaded hydropower generation until the storage returns to the upper limit.

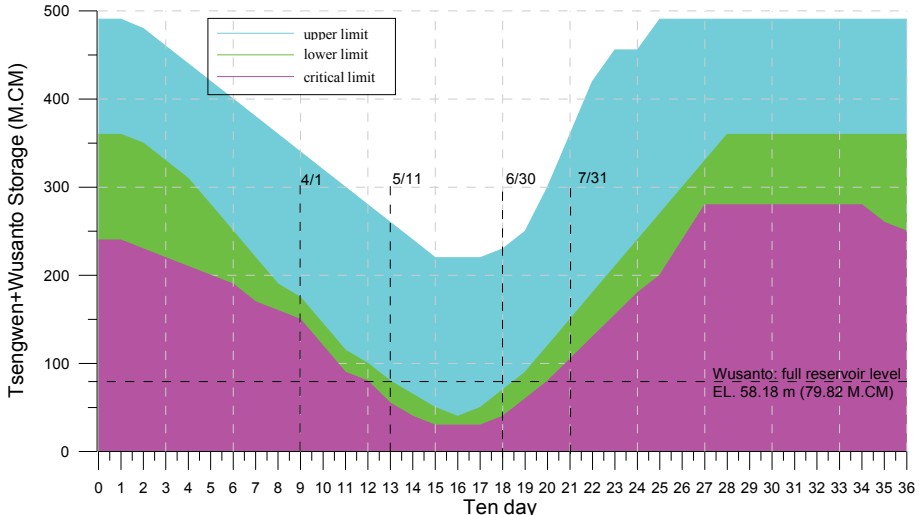

**Fig. 2 Joint operating rule curves of the Tsengwen and Wushanto Reservoirs**

2. Distributing storage to individual reservoirs: Based on the calculated total water supply, the total end-of-period storage in the system can be estimated with the expected reservoir inflow during one single operating period. The release from each individual reservoir can then be determined by applying an individual reservoir storage balancing function, such as storage balancing curves. Fig. 3 exhibits the storage balancing curves for the Tsengwen and Wushanto Reservoirs in early April (SRWRO, 2012). The horizontal axis in the figure



measures the total storage in the system, and the two curves represent the suggested target

storages for the respective reservoirs with regard to various total storage amounts. These

curves vary during each ten-day period within a year to facilitate efficient storage

allocation according to the pattern of water demands and reservoir inflow.

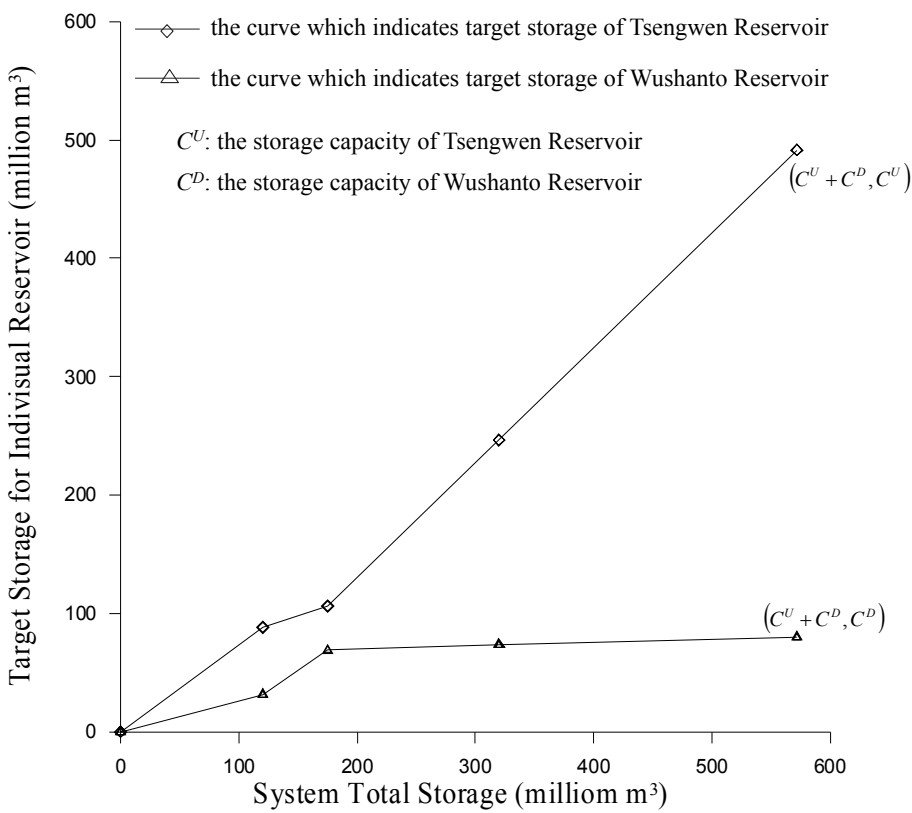

**Fig. 3 Storage balancing curves for Tsengwen and Wushanto Reservoirs in the tenth**

**ten-day period (early April)**

The first part of the proposed method requires appropriate adjustment of the storage

10   balancing curves before and during the periods feasible for empty flushing. This adjustment

would prioritize the water released from the primary reservoir while preserving storage in the

other. This complies with the aforementioned QCWS1 and QCFS2, and creates a favorable



initial condition for empty flushing.

### 2.2.2 Conditions for initiation of an empty flushing operation

Water supply simulation of historical daily reservoir inflow records is sequentially performed according to the joint operating rules. During the simulation, empty flushing operation is activated when all of the following conditions are satisfied:

1. The current simulating date falls within the pre-evaluated feasible timeframe for empty flushing.

2. The storage of the primary reservoir is lower than a threshold $T^U$. This ensures the satisfaction of QCFS2. Theoretically, a higher value of $T^U$ allows for the initiation of drawdown flushing at higher primary reservoir storage levels, thus increasing the range of opportunities for empty flushing. Nonetheless, a higher $T^U$ incurs the risk that, if subsequent reservoir inflow falls short of predicted values, the emptied storage may not be replenished.

3. The total storage in the backup reservoirs is greater than a threshold $T^D$. This ensures meeting QCWS1. A higher value of $T^D$ elevates the stability of water supply during empty flushing. In cases where either this or the above condition has not been met, demand should be supplied from the storage of primary reservoir as much as possible, or storage should be diverted from the primary reservoir to the others. However, this storage reallocation may be limited by the water transmitting capacities between reservoirs. Such that the conditions for initiating empty flushing may not be met within the pre-specified feasible period for flushing. Therefore, a higher $T^D$ may reduce the opportunities to perform empty flushing.





### 2.2.3 Estimation of the flushed sediment discharge

Once the activation conditions are met, the gates of the bottom outlets of the primary reservoir are fully opened to empty the storage and route the inflowing water and sediments. The release from the primary reservoir may cause blockages of the downstream water diversion or water treatment facilities due to its high sediment concentration. Thus the water supply may rely solely on the storage preserved in the other reservoirs. During empty flushing, the empirical formula developed by the International Research and Training Center on Erosion and Sediment (IRTCES) in Tsinghua University, Beijing (IRTCES, 1985) is employed for the estimation of releasing sediment discharge from the primary reservoir. The formula is based on measurements from 14 reservoirs in China:

$$QC_t = \psi \frac{Q_t^{1.6} S_f^{1.2}}{W^{0.6}}$$

(1)

where $QC_t$ and $Q_t$ denote the sediment discharge (T/s) and water discharge (m³/s) flushed from the primary reservoir during the t-th simulating day, respectively; $S_f$ represents the energy slope associated with the flow in the primary reservoir during empty flushing; $W$ is the width of the flushing channel (m), which can be estimated using the empirical formula $W = 12.8 \cdot Q^{0.5}$ (Atkinson, 1996), and $\psi$ is the flushing coefficient, associated with the characteristics of the sediment and topography of the reservoir.

### 2.2.4 Conditions for termination of empty flushing operation

Empty flushing operations should be terminated if either of the following circumstances occurs:

1. The flushing should be terminated when the flood flow has raised the WSL of the primary reservoir and inflow subsequently recedes to be below the capacity of associated bottom




outlets. This situation indicates that the operation has been successfully timed to encounter a flood and should thus be ended when the flood ends.

2. The flushing should be ended when the storage of backup reservoirs decreases to below a threshold $T^d$. This condition prevents short-term water shortages following flushing operations resulting from insufficient storage. During the flushing operation, the primary reservoir will remain empty in the absence of floods, so providing water supply will gradually reduce available storage in the other reservoirs. A higher value of $T^d$ will cause the storage below threshold more quickly, thus reducing the window of operation for empty flushing. Nonetheless, adequate reservoir inflow and proper storage reallocation after an earlier termination of one flushing operation will facilitate the re-initiation of a subsequent operation during the feasible period for empty flushing. Thus, under conditions of a higher $T^d$ value, the pattern of empty flushing may be transformed from a few operations of longer duration into multiple intermittent operations of shorter durations.

A generalized water allocation simulation model (GWASIM) developed by Chou and Wu (2010) is used to simulate the alternating operations of empty flushing and joint water supply according to the aforementioned rules and conditions. The structure of this model is formulated in network flow programming. It has already been implemented in the planning and management of all major water resources systems in Taiwan. Details of its simulations regarding the operations of multi-reservoir systems, such as those in Subsection 3.1, can be found in Chou et al. (2006) and Chou and Wu (2014).

**2.2.5 Evaluation of optimal empty flushing strategies**

The storage thresholds for activating and terminating an empty flushing operation as described in subsections 2.2.2 and 2.2.4 are regarded as parameters. These parameters are



calibrated to maximize the total desilting volume without inducing intolerable water shortage scenarios. Since empty flushing is restricted to a feasible period suggested to span from the end of the dry season to the early wet season, the occurrence of a subsequent flood which may cause reservoir spillage will fully compensate for the impact of probable water shortage after empty flushing. Thus the incremental shortage will be concentrated in a few months following the feasible flushing period, during each of which the monthly shortage increment and ratio is calculated respectively:

$$d_{n,m}^{I} = d_{n,m} - d_{n,m}^{0}, \quad m = 0,1,2,...,n^{m}, \quad n = 1,2,...,n^{y} \tag{2}$$

$$d_{n,m}^{R} = \frac{d_{n,m}}{D_{m}}, \quad m = 0,1,...,n^{m}, \quad n = 1,2,...,n^{y} \tag{3}$$

where $d_{n,m}^{I}$ and $d_{n,m}^{R}$ represent the water shortage increment and ratio during the $m$-th month following the feasible period of empty flushing in the $n$-th simulating year; $D_{m}$ denotes the water demand during the $m$-th month following empty flushing; $d_{n,m}$ and $d_{n,m}^{0}$ represent simulated water shortages under conditions with and without empty flushing operations, $n^{m}$ is the number of months within which the impact of empty flushing on water supply is carried over; and $n^{y}$ is the number of simulating years. The formulation of the optimization problem is as follows:

$$Maximize \quad \sum_{t=1}^{n^{t}} QC_{t} \tag{4}$$

subject to

$$\max_{n=1,...,n^{y}} (d_{n,m}^{R} \mid d_{n,m}^{I} > 0) \leq \alpha \quad m = 0,1,2,...,n^{m} \tag{5}$$





where $n^t$ is the total number of days within the simulating horizon; $QC_t$ is the simulated sediment discharge from the primary reservoir by empty flushing on the t-th day. It is determined by substituting the release of the primary reservoir during the flushing period into Eq. (1), and $\alpha$ is the maximum acceptable monthly water shortage ratio induced by empty

flushing. The BOBYQA, a nonlinear optimization algorithm of Powell (2009), is used to solve the problem. The details of BOBYQA can be found in Powell (2009) and the barrier function approach to handle the constraint of Eq. (5) can be found in Chou and Wu (2015).

### 2.3 Case study and experimental setup

The joint operating system of the Tsengwen and Wushanto Reservoirs in southern

Taiwan is selected for case study. Fig. 4 shows the location of these reservoirs. The Tsengwen Reservoir is located in the upper section of the Tsengwen River, with a watershed area of 481.6 km$^2$. The original effective capacity with the WSL as the normal pool level El. 227 m was 631.2 million m$^3$ when the reservoir was completed in 1973. Operated by the SRWRO, its purposes include agricultural and public water supply, flood control, and hydropower

generation. The annual inflow volume of the reservoir from 1974 to 2013 is 116.7 million m$^3$ and the annual inflowing sediment volume is estimated as 5.6 million m$^3$ by the SRWRO.

Located 6 km downriver of the Tsengwen Reservoir, the East Weir diverts the releases from the Tsengwen Reservoir to the Wushan Hill Tunnel at a conveyance capacity of 56 m$^3$/s. The water is conveyed 3.3 Km to the West Weir on the Guantien Creek and then flows into

the Wushanto Reservoir.

The Wushanto Reservoir is situated to the southwest of the Tsengwen Reservoir in the upper section of Guantien Creek, a tributary of Tsengwen River. The watershed area of the Wushanto Reservoir is only 60 km$^2$, which renders it conceptually off-stream. In 2011, its effective capacity was measured at 79.82 million m$^3$. The Chianan Irrigation Association

manages the Wushanto Reservoir in coordination with the release from Tsengwen Reservoir,





supplying irrigation water to over 70 thousand hectares of farmland in the Chianan Plain and providing the public and industrial water to the Tainan City and a portion of Chiayi City and Chiayi County. The joint operating rule curves for the Tsengwen and Wushanto Reservoirs are presented in Fig. 1 of Subsection 2.2.1.

From the beginning of operations in April 1973 until September 2013, the effective capacity of the Tsengwen Reservoir was reduced from 631.2 million $m^3$ to 473.3 million $m^3$. A major cause was Typhoon Morakot in 2009, which brought record-breaking rainfall to the reservoir watershed. The flood inflow of Tsengwen Reservoir peaked at 11,729 $m^3$/s, which is only slightly below the peak of its probable maximum flood as 12,430 $m^3$/s. Measurements at the end of 2009 indicated that the sedimentation of Tsengwen reservoir had increased by a massive 91.08 million $m^3$ that year, which is 19.7 times that of the average annual sedimentation between 1973 and 2008. In response to the substantial increase in sedimentation, the SRWRO improved the permanent river outlets (PRO) of the Tsengwen Reservoir to promote desilting. The improvements include changing the original Howell-Bunger valve to a jet flow gate and increasing the releasing capacity to 177 $m^3$/s to facilitate more effective hydraulic sediment venting during floods. At present, the elevation of bed in front of the dam of Tsengwen Reservoir have reached 171 El. m. This level is higher than the bottom of the inlets of the PRO at the 153.37 El. m, which allows empty flushing through this newly updated outlet.

The process of deriving the optimal empty flushing strategy starts from determining the feasible period for implementing empty flushing to the Tsegnwen Reservoir, based on the criteria mentioned in subsections 2.1 and 2.2. The storage balancing curves for these two reservoirs are then modified before and during the evaluated feasible period to create favorable conditions to initiate empty flushing. Different storage thresholds for activation and termination of an empty flushing operation are then tested to preliminarily assess the trade-off





between desiltation and water supply. These storage thresholds are calibrated to maximize the

desilting volume without inducing intolerable water shortage. Finally, validation analysis and

economic evaluation are performed to verify the effectiveness of the derived strategy. Details

of this process and the results are presented and discussed in the following section.

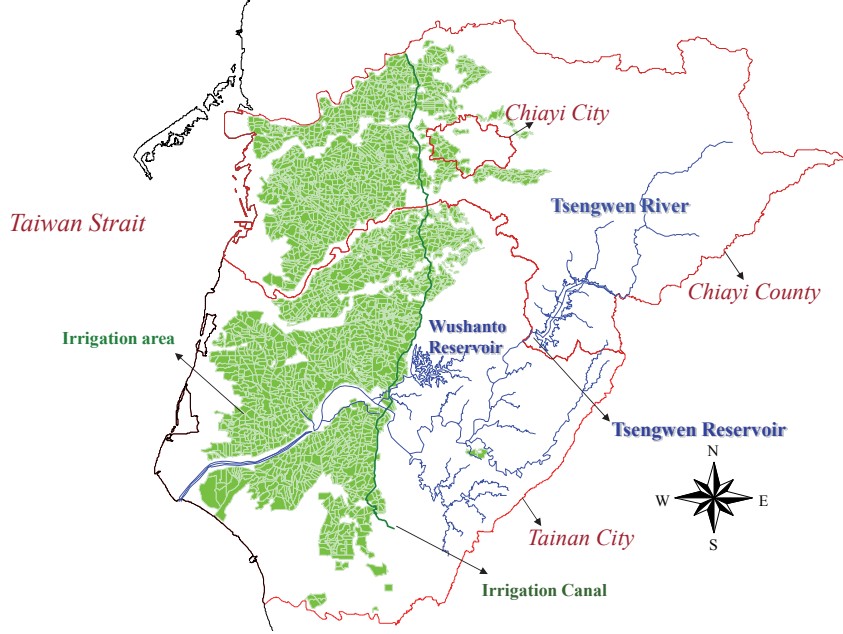

**Fig. 4 The map of Tsengwen and Wushanto Reservoirs**

## 3. Result and discussion

### 3.1 Determination of the feasible period for empty flushing

Fig. 5 illustrates both water demand of this system and average inflow to the Tsengwen

Reservoir in ten-day increments over a year. As can be seen, inflow to the reservoir generally

begins increasing between late May and early June, as precipitation rises during the beginning

of the wet season. This is also the period in which the irrigational water demand, which

constitutes the majority of total demands, is lower. The first semiannual rice crop is harvested,





and the second semiannual irrigation just begins. As shown in Fig. 2, between May 11 and June 30, the lower limit of the operating rule curves is below the effective capacity of the Wushanto Reservoir. Even if the Tsengwen Reservoir is empty, as long as the Wushanto Reservoir is full, the total storage of the system would still exceed the lower limit of the operating rule curves, such that the demand for water could be satisfied. All of these characteristics indicate that the climate and operating conditions during May and June are favorable for empty flushing of Tsengwen Reservoir.

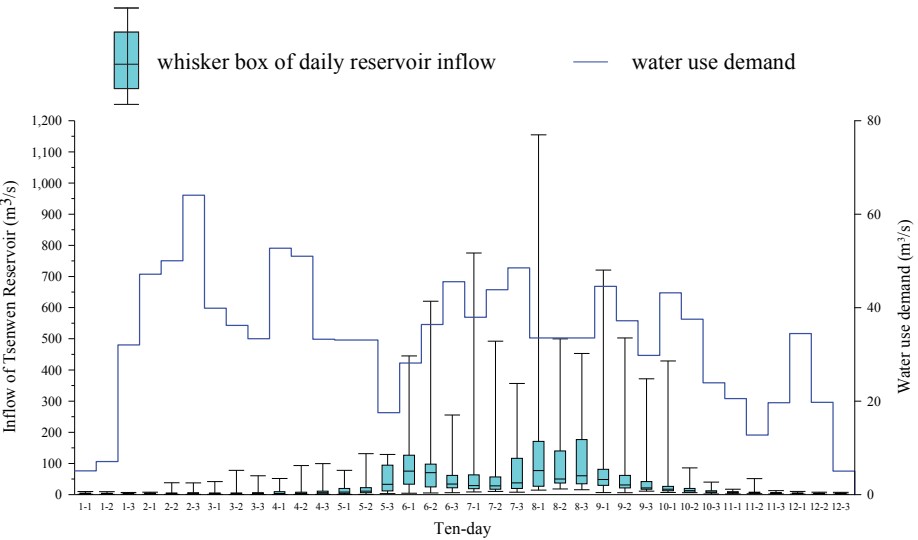

**Fig. 5 Demand and inflow patterns of Tsengwen Reservoir during ten-day periods throughout the year**

To validate the above assertion, sequential water supply simulation is performed using the daily inflow records of the reservoirs from 1975 to 2009 and the joint operating rules as described in subsection 2.2.1 in the absence of flushing operations. Fig. 6 illustrates the network of the water resources system. The simulated results provide a basis for calculating the probability that the storage in the Tsengwen Reservoir drops below 20 million m$^3$ for





preparing empty flushing timely in a given month while the Wushanto Reservoir storage simultaneously exceeds the lower limit of the operating rule curves. The results are displayed in the second rows of Tables 2 and 3. The results show that in May, there is a 52 % probability that the storage of the Tsengwen Reservoir will drop below 20 million m$^3$ and an 8 %

5 probability that the Tsengwen Reservoir storage drops below 20 million m$^3$ while the Wushanto Reservoir storage simultaneously exceeds the lower limit. In June, the two probabilities are 31 % and 14 %, respectively. These two months present the highest probabilities during a year. The respective storage levels of the Tsengwen and Wushanto Reservoirs each satisfy the abovementioned conditions only between May 11 and June 20,

10 which is thus selected as the feasible period for empty flushing in the Tsengwen Reservoir.



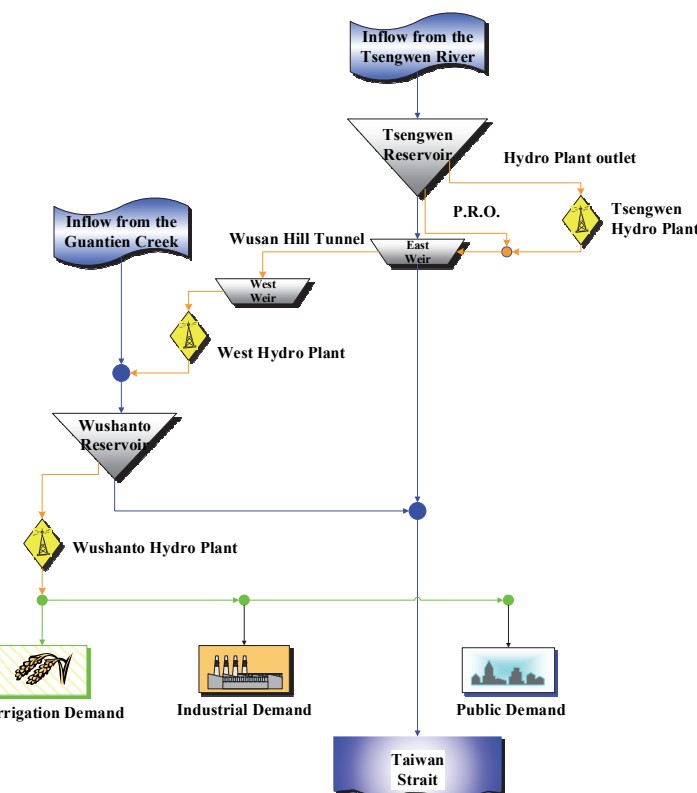

**Fig. 6 Network of the joint operating system of Tsengwen and Wushanto Reservoirs**

**3.2 Schemes for the modification of storage balancing curves**

5       In order to maximize opportunities for empty flushing of Tsengwen Reservoir, storage

balancing curves such as Fig. 3 are modified to preserve as much storage in the Wushanto

Reservoir as possible by satisfying demands first with Tsengwen Reservoir storage. Fig. 7

depicts the modified balancing curves. Based on the simulation conditions given in subsection

3.1, three additional simulations are conducted in which the modified storage balancing

10    curves are applied during (1) May 1 to June 20, (2) April 1 to June 20, and (3) March 1 to

June 20. For the probabilities of the storage in the Tsengwen Reservoir dropping below 20

million $m^3$ and those of the storage in the Tsengwen Reservoir dropping below 20 million $m^3$



in conjunction with the storage in the Wushanto Reservoir being higher than the lower limit, we adopted the values presented in Rows 3 and 4 of Tables 2 and 3. Table 4 presents the monthly average water shortage ratio resulting from the original and modified storage balancing curves. The results demonstrate that the probability of favorable storage

5    distribution for empty flushing during May and June can be effectively elevated by modifying storage balancing curves in April. Furthermore, the water shortage ratios generated by the modified balancing curves are no more than 0.01 higher than those from the original balancing curves, which means that the modification has only a trivial impact on the water supply. The results also indicate that the average water shortage ratio during the wet season

10    drops considerably after July. This is because the first typhoon of the wet season generally occurs in July or early August, bringing substantial inflow to the reservoirs. Thus in the following evaluation of empty flushing strategies, only the water shortage scenarios through the end of July are selected to represent the impact of empty flushing on water supply.





**Table 2 Monthly probabilities of Tsengwen Reservoir storage dropping below 20 million**

**m$^3$ under various strategies of storage allocation**

| Strategy \ Month | Jan | Feb | Mar | Apr | May | June | Jul | Aug | Sep | Oct | Nov | Dec |
|---|---|---|---|---|---|---|---|---|---|---|---|---|
| Original storage balancing curves | 0.01 | 0.04 | 0.13 | 0.13 | 0.52 | 0.31 | 0.12 | 0.02 | 0.00 | 0.00 | 0.00 | 0.00 |
| Modified curves from May to June | 0.03 | 0.04 | 0.13 | 0.13 | 0.64 | 0.33 | 0.13 | 0.03 | 0.00 | 0.00 | 0.00 | 0.01 |
| Modified curves from April to June | 0.03 | 0.04 | 0.13 | 0.34 | 0.78 | 0.33 | 0.13 | 0.02 | 0.00 | 0.00 | 0.00 | 0.01 |
| Modified curves from March to June | 0.03 | 0.04 | 0.13 | 0.45 | 0.78 | 0.33 | 0.13 | 0.02 | 0.00 | 0.00 | 0.00 | 0.01 |





**Table 3 Monthly probabilities of storage in Tsengwen Reservoir dropping below 20 million m³ with storage in Wushanto Reservoir exceeding the lower limit under various strategies of storage allocation**

| Month Strategy | Jan | Feb | Mar | Apr | May | June | Jul | Aug | Sep | Oct | Nov | Dec |
|---|---|---|---|---|---|---|---|---|---|---|---|---|
| Original storage balancing curves | 0.00 | 0.00 | 0.00 | 0.00 | 0.08 | 0.14 | 0.00 | 0.00 | 0.00 | 0.00 | 0.00 | 0.00 |
| Modified curves from May to June | 0.00 | 0.00 | 0.00 | 0.00 | 0.11 | 0.16 | 0.00 | 0.00 | 0.00 | 0.00 | 0.00 | 0.00 |
| Modified curves from April to June | 0.00 | 0.00 | 0.00 | 0.00 | 0.14 | 0.16 | 0.00 | 0.00 | 0.00 | 0.00 | 0.00 | 0.00 |
| Modified curves from March to June | 0.00 | 0.00 | 0.00 | 0.01 | 0.14 | 0.16 | 0.00 | 0.00 | 0.00 | 0.00 | 0.00 | 0.00 |



**Table 4 Average monthly water shortage ratio simulated under various strategies of storage allocation**

| Month<br>Strategy | Jan | Feb | Mar | Apr | May | June | Jul | Aug | Sep | Oct | Nov | Dec |
|---|---|---|---|---|---|---|---|---|---|---|---|---|
| Original storage balancing curves | 0.10 | 0.17 | 0.23 | 0.29 | 0.22 | 0.07 | 0.10 | 0.06 | 0.06 | 0.09 | 0.09 | 0.11 |
| Modified curves from May to June | 0.11 | 0.18 | 0.24 | 0.30 | 0.23 | 0.07 | 0.10 | 0.06 | 0.07 | 0.09 | 0.09 | 0.12 |
| Modified curves from April to June | 0.11 | 0.18 | 0.24 | 0.31 | 0.23 | 0.07 | 0.10 | 0.06 | 0.07 | 0.09 | 0.09 | 0.12 |
| Modified curves from March to June | 0.11 | 0.18 | 0.24 | 0.31 | 0.23 | 0.07 | 0.10 | 0.06 | 0.07 | 0.09 | 0.09 | 0.12 |



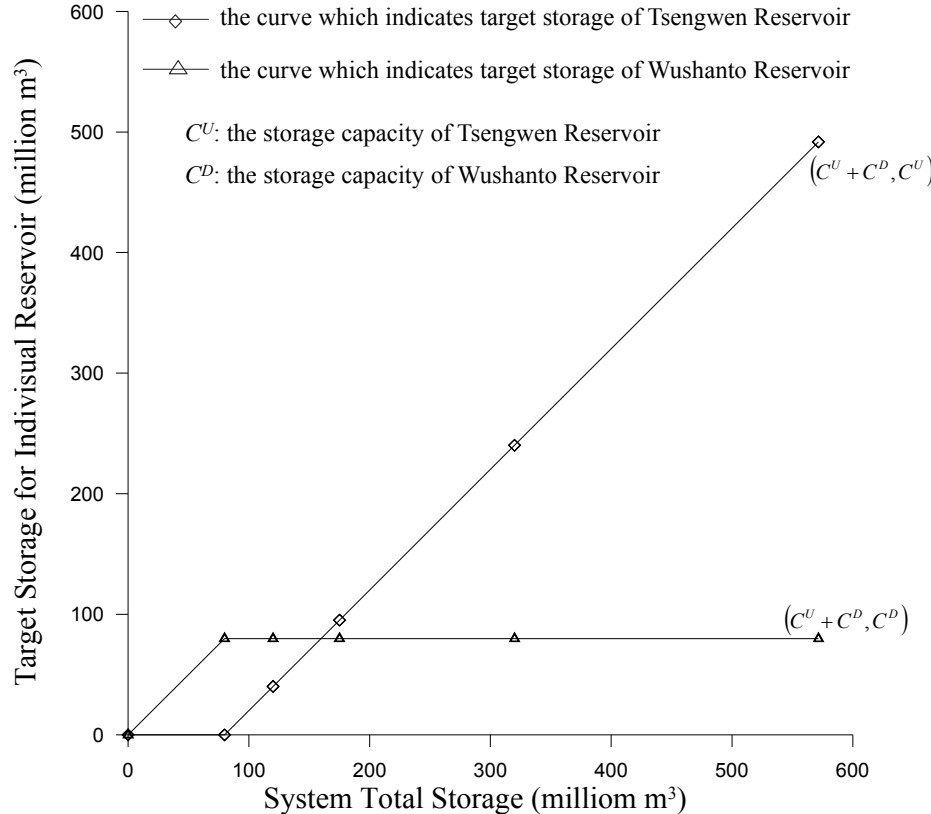

**Fig. 7 Modified storage balancing curves using Tsengwen Reservoir as primary source to satisfy demand for water**

### 3.3 Preliminary simulations and assessment of empty flushing strategies

The storage thresholds to activate and terminate an empty flushing operation, i.e. $T^U$ for the Tsengwen Reservoir, and $T^D$ and $T^d$ for the Wushanto Reservoir, are regarded as parameters to be optimized. These parameters are allowed to vary during different ten-day periods from May 11 to June 20 in order to promote the performance of desilting and backup water supply. Before actually optimizing these parameters, preliminary simulations are performed with constant storage thresholds throughout May 11 to June 20. This process facilitates determination of a good initial solution as well as a basis for comparison to





measure the effects of optimization. The preliminary simulations consider seven different $T^U$ values, ranging from 0 to 60 million m$^3$ with a constant interval of 10 million m$^3$, for the Tsengwen Reservoir. Six values (including 55, 60, 65, 70, 75 and 79 million m$^3$) for $T^D$ and nine values (including 30 to 70 million m$^3$ with a constant interval as 5 million m$^3$) for $T^d$ are

5 considered. These values contribute to a total of 308 combinations of empty flushing strategies in which $T^d$ is less than $T^D$.

  To determine the volume of flushed sediments, measurements of sediment concentration from the PRO release of the Tsengwen Reservoir are used to establish the relationship between the flushing coefficient $\psi$ and the WSL of the reservoir, as shown in

10 Fig. 8. It reveals that $\psi$ approaches a fixed value of 2.5 when the WSL in the Tsengwen Reservoir exceeds 190 El. m. Measurements of the effluent sediment concentration at lower WSLs are not available currently; therefore, we referred to Atkinson (1996), who suggested using $\psi = 60$ when the capacity of bottom outlets is limited. Atkinson (1996) also suggested that when the water depth of a reservoir exceeds 30 % its maximum depth, the flushing

15 efficiency will decrease significantly. The 30 % depth of the Tsengwen Reservoir is approximately at the elevation of 185 El. m, which corresponds to an impoundment of 21.81 million m$^3$. To prevent overestimating the effectiveness of empty flushing, it is assumed that if a flood during an empty flushing operation raises the WSL of the Tsengwen Reservoir to exceed 185 El. m, then the flushed sediment volume from the PRO is set to be 0. In addition,

20 the assumption of uniform flow condition during empty flushing allows the use of thalweg slope, which is 0.0032 according to the measurement in 2011, to represent the energy slope as required in Eq. (1). Then, according to the simulated PRO release during the empty flushing operation, the flushed sediment discharge as well as the desilting volume during the simulating time horizon can be estimated using Eq. (1).




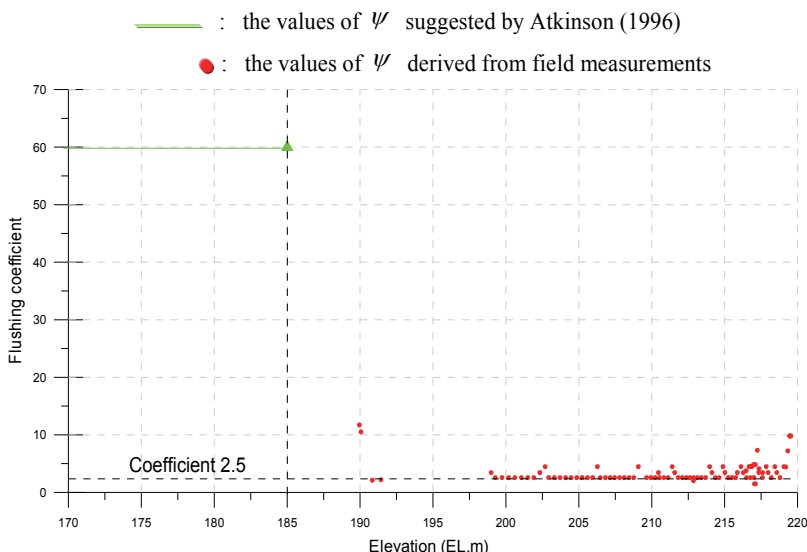

**Fig. 8 Relationship between the flushing coefficient and WSL of Tsengwen Reservoir**

To assess the impact on water supply following empty flushing, the ratio and increments of water shortages during the remaining periods of June and during the entire July in each simulated year are calculated. The maximum monthly water shortage ratio is then calculated according to the following equations:

$$d^R_{\max,0} = \max_{n=1,\dots,n^y} (d^R_{n,0} \,|\, d^I_{n,0} > 0) \tag{6}$$

$$d^R_{\max,1} = \max_{n=1,\dots,n^y} (d^R_{n,1} \,|\, d^I_{n,1} > 0) \tag{7}$$

$$d^R_{\max} = \max (d^R_{\max,0},\ d^R_{\max,1}) \tag{8}$$

where $d^R_{n,0}$ and $d^I_{n,0}$ denote the ratio and increments of water shortage between the end of the last empty flushing operation to the end of June in the $n^{th}$ simulated year, while $d^R_{n,1}$ and $d^I_{n,1}$

are the ratio and increments of shortage throughout July in the $n^{th}$ year; $d_{max,\,0}^{R}$ and $d_{max,\,1}^{R}$ represent the maximum water shortage ratio during the period between the completion of the last flushing operation until the end of June and during July, respectively. The primary causes of water shortage included the absence of heavy rainfall during May and June and a delayed

arrival of the first typhoon in July. These conditions lead to insufficient inflow during June and July, which necessitate water rationing according to the joint operating rule curves.

According to the above conditions, simulations of the 308 combinations are performed using the original storage balancing curves. The resulting average annual desilting volume and maximum monthly water shortage ratio induced by empty flushing are then calculated.

The results are presented in Fig. 9. The simulations are then repeated by applying the modified storage balancing curves in Fig. 7 to the period between April and June, the results of which are displayed in Fig. 10. A comparison of Figs. 9 and 10 shows that the modified storage balancing curves effectively enhance the effectiveness of desilting. For instance, strategies with $d_{max}^{R}$ between 0.17 and 0.23 correspond to a maximum annual desilting

volume of 0.06 million m³/year in Fig. 9, whereas the same strategies in Fig. 10 result in an increase of desilting volume reaching 0.54 million m³/year.





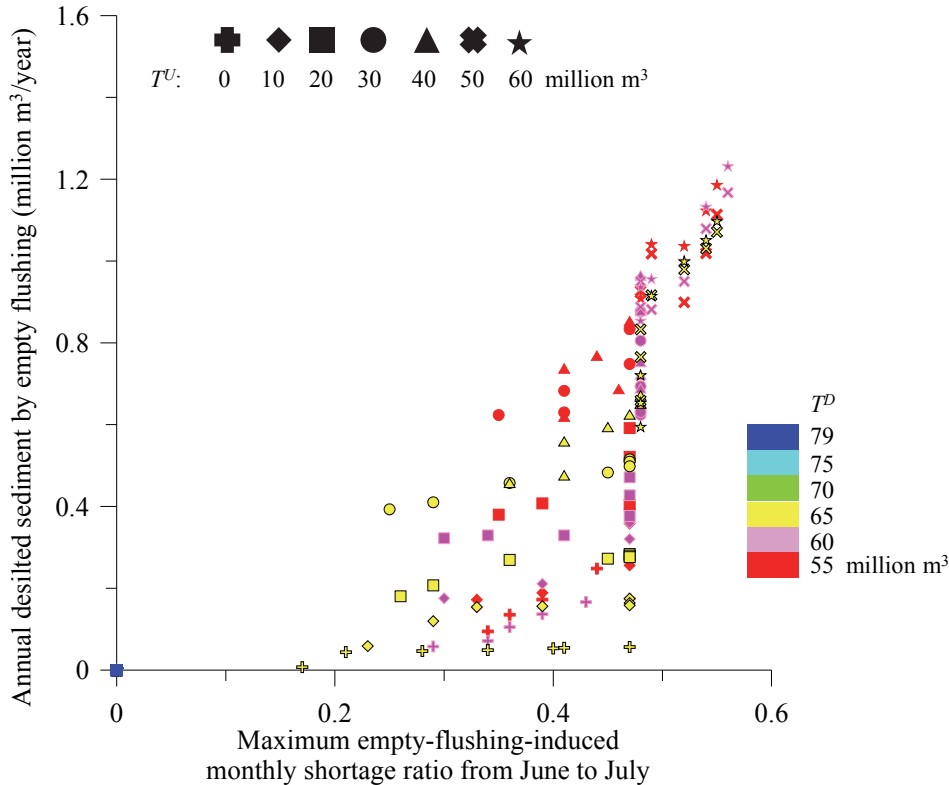

**Fig. 9 Simulation results of various empty flushing strategies using the original storage**

**balancing curves**





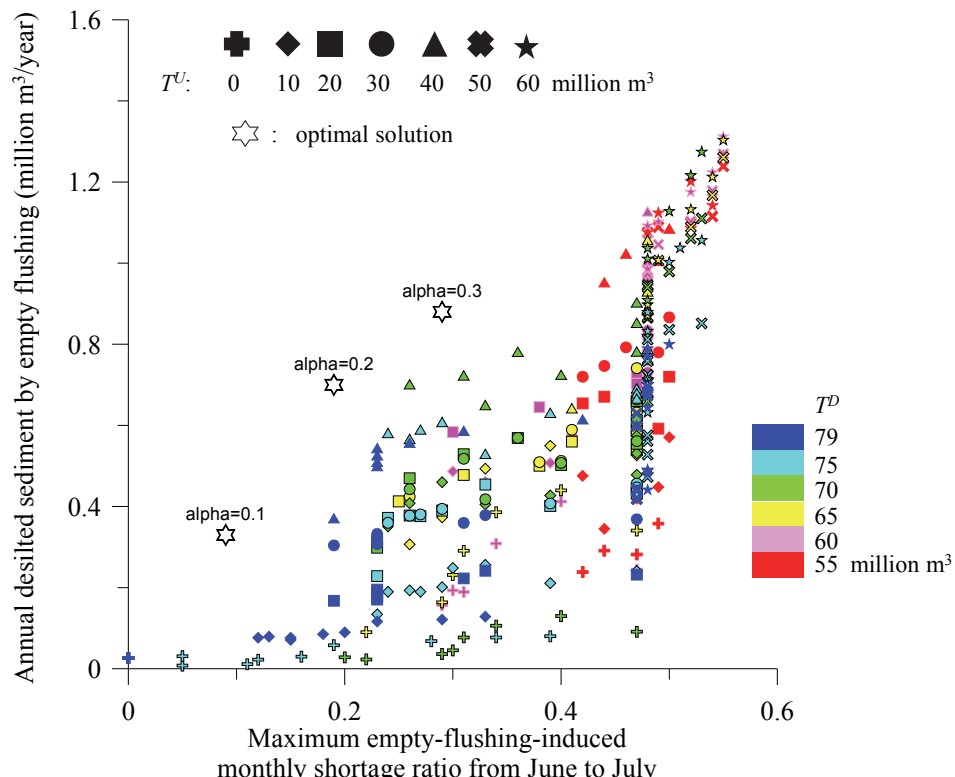

**Fig. 10 Simulation results of various empty flushing strategies using modified storage balancing curves from April to June**

### 3.4 Optimization of empty flushing strategies

According to the maximum monthly shortage ratio adopted in subsection 3.3, the constraint of Eq. (5) is divided as follows to enable greater precision in control over water shortages induced by empty flushing:

$$d_{max,0}^R = \max_{n=1,...,n^y}(d_{n,0}^R \,|\, d_{n,0}^I > 0) \le \alpha \qquad (9)$$



$$d_{\max,1}^{R} = \max_{n=1,\dots,n^{y}} (d_{n,1}^{R} \big| d_{n,1}^{I} > 0) \leq \alpha \qquad (10)$$

Coupling the established simulation model with the optimization algorithm leads to the optimal solution of Eqs. (4), (9) and (10) under a specific value of the maximum acceptable monthly shortage ratio, $\alpha$. In the case study, three values of $\alpha$, including 0.1, 0.2 and 0.3, are tested. The corresponding optimal storage thresholds to activate and terminate an empty flushing operation are presented in Table 5. The average annual desilting volume and maximum monthly shortage ratio induced by empty flushing are also marked in Fig. 10. Due to the frequency of drought in this system, the optimal strategy associated with $\alpha$ =0.1 is selected due to its conservativeness. Table 6 displays the simulated events of empty flushing based on this calibrated strategy.

Table 6 presents the monthly shortage ratio in July following the empty flushing operations in 1989 and 2009 both reach 0.41. However, the corresponding shortage increments are both 0; therefore, they did not violate the constraints of Eqs. (9) and (10). Figs. 11 and 12 present the hydrographs of inflow to the Tsengwen Reservoir and the total system storage during these two years. As shown in Table 6 and these figures, the Tsengwen Reservoir is nearly empty and the Wushanto Reservoir is nearly full before the initiation of empty flushing operations. Thus, the empty flushing operations only consume the inflow of Tsengwen Reservoir during a 2 to 3 days period. These water consumption volumes are too insignificant to induce the subsequent water shortage seen in July. The primary reason for the subsequent shortage is the delayed arrival of the first typhoon in late July or early August, by which time the total storage falls below the critical limit of the joint operating rule curves and water rationing is applied. Following the arrival of the typhoons, however, the total reservoir storage exceeds the lower limit and even the upper limit of rule curves, thereby alleviating the water shortage.



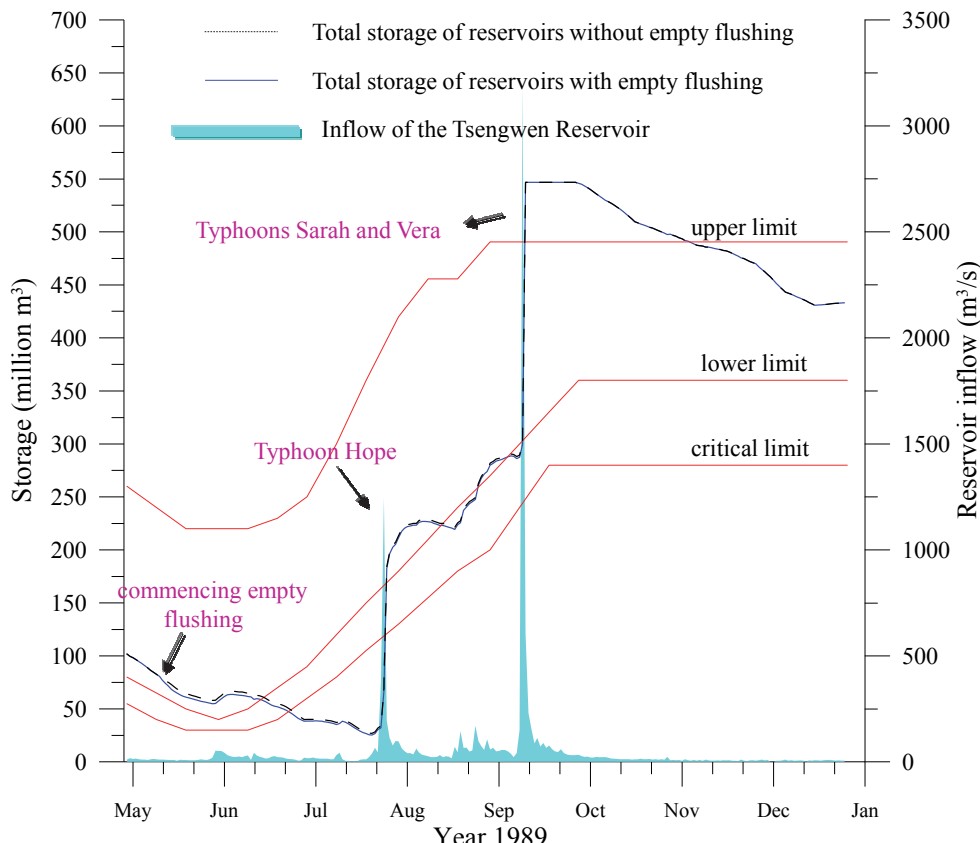

**Fig. 11 The hygrographs of Twengwen Reservoir inflow and total system storage**

**throughout 1989**





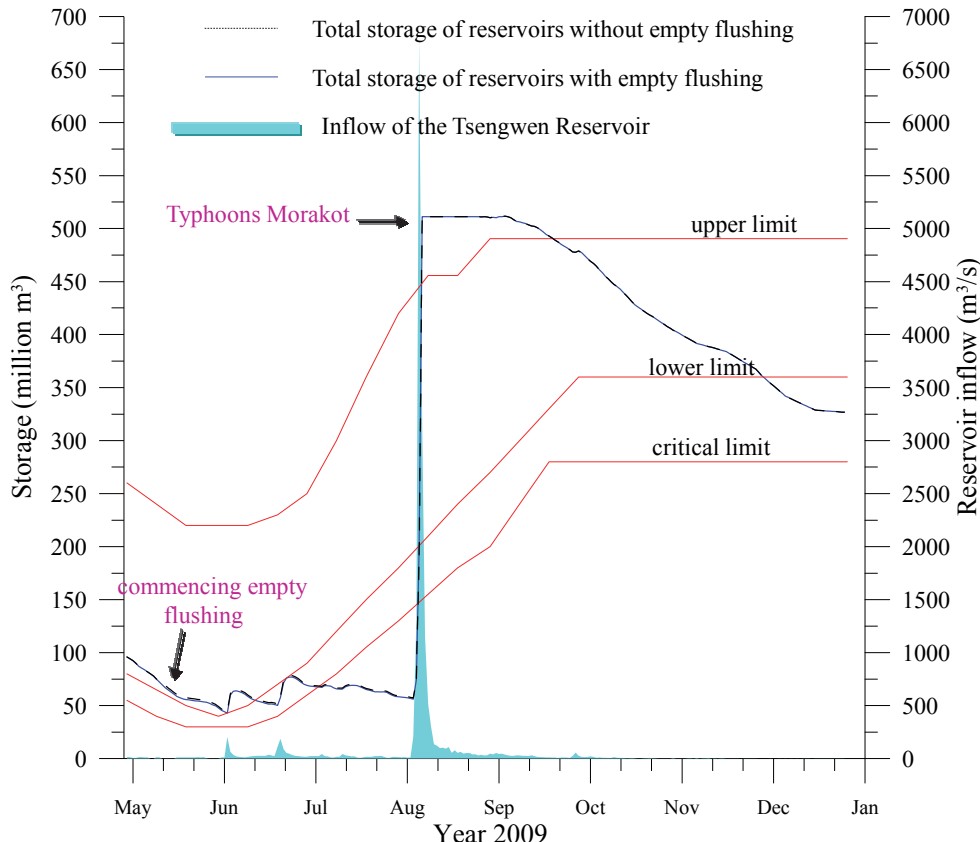

**Fig. 12 The hygrographs of Twengwen Reservoir inflow and total system storage throughout 2009**

### 3.5 Validation analysis of the optimal strategies

5    The optimal strategies in Table 6 are derived by linking the optimization algorithm to the model that simulates operations according to the records of daily reservoir inflow between 1975 and 2009. Following this calibration period, the records through the end of 2013 are used to verify the effectiveness of the established strategy. The results of the validating simulation indicate that two flushing operations could have been conducted during this period,

10   one in 2010 and the other in 2013. Table 7 summarizes the two events. Figs. 13 and 14, respectively, present the hydrographs of reservoir inflow and total system storage from May





to December of these years. Clearly, following the initiation of empty flushing operations in early June of 2010, the monthly water shortage ratio during July is 0.18, which is higher than the 0.12 that would have been the case without empty flushing. The increased shortage ratio is induced by drawdown and empty flushing, which cause the total storage to fall below the

critical limit earlier in July. Empty flushing thus necessitates a longer water rationing period. Nonetheless, torrential rains in late July elevate the storage to exceed the lower limit, thereby resolving the shortage crisis. The major impact of water shortage during this period is on the second semiannual irrigation operation, which requires large quantities of water during July. One of the mitigation measures is to postpone the beginning irrigation schedule no later than

August 10. For example, in May and June of 2004, the total storage in the Tsengwen and Wushanto Reservoirs fell below the critical limit, which delayed the second semiannual irrigation from the originally planned June 6 to July 17 when Typhoon Mindulle invaded and elevated the storage above the upper limit in early July. For the empty flushing operation from May 11 to May 13 in 2013, though, no water shortage occurs during June and July.

This section is concluded with an economic evaluation of the optimized empty flushing strategy. With the strategy associated with $\alpha = 0.1$, the average annual desilting volume is 0.324 million m$^3$/year covering the calibration and validation periods, while the induced shortage increments are 0.821 million m$^3$/year. The SRWRO once estimated that dredging 1 m$^3$ of sediment from the Tsengwen Reservoir will cost NTD\$524. On the other hand, the

contract between the SRWRO and Chianan Irrigation Association stipulates that transferring 1 m$^3$ of agricultural water to public supply will cost NTD\$4.248 for enhancing irrigation management. Accordingly, the net annual benefits of the suggested empty flushing strategy will be approximately 166 million NTD, which shows its economic superiority over dredging.





**Table 5 Optimal empty flushing strategies based on acceptable water shortage rates following the completion of empty flushing operations**

| Ten-day period | Tsengwen Reservoir flushing initiation condition $T^U$ (million m³) | | | Wushanto Reservoir flushing initiation condition $T^D$ (million m³) | | | | Wushanto Reservoir flushing termination condition $T^d$ (million m³) | | | | Water shortage increment until July (million m³ /year) | Annual desilting volume (million m³/year) |
|---|---|---|---|---|---|---|---|---|---|---|---|---|---|
| | 14th | 15th | 16th | 17th | 14th | 15th | 16th | 17th | 14th | 15th | 16th | 17th | | |
| $\alpha = 0.1$ | 2.0 | 40.0 | 40.0 | 0.0 | 75.0 | 70.0 | 68.0 | 79.8 | 70.0 | 51.0 | 60.0 | 76.0 | 0.73 | 0.33 |
| $\alpha = 0.2$ | 60.0 | 40.0 | 44.0 | 30.0 | 75.0 | 61.0 | 67.5 | 79.8 | 55.0 | 40.0 | 44.0 | 60.0 | 2.31 | 0.70 |
| $\alpha = 0.3$ | 80.0 | 40.0 | 40.0 | 30.0 | 75.0 | 57.0 | 67.5 | 79.8 | 51.1 | 40.0 | 39.2 | 60.0 | 3.25 | 0.88 |



**Table 6 Simulated empty flushing events based on optimal strategy with $\alpha = 0.1$**

| Year | Initiation date | Storage in Tsengwen Reservoir at initiation (M. m³) | Storage in Wushanto Reservoir at initiation (M. m³) | Termination date | Storage in Tsengwen Reservoir at termination (M. m³) | Storage in Wushanto Reservoir at termination (M. m³) | Water shortage increment until June (M. m³) | Water shortage ratio until June | Water shortage increment in July (M. m³) | Water shortage ratio in July | Desilting volume (M. m³) |
|------|------|------|------|------|------|------|------|------|------|------|------|
| 1977 | 6/05 | 0 | 68.27 | 6/08 | 0 | 59.17 | 0.00 | 0.00 | 0.00 | 0.00 | 0.13 |
| 1984 | 5/11 | 18.64 | 79.82 | 5/14 | 0 | 68.74 | 4.31 | 0.04 | 1.71 | 0.01 | 0.06 |
| 1984 | 5/23 | 0 | 71.53 | 6/01 | 42.68 | 66.18 |  | 0.04 |  | 0.02 | 1.06 |
| 1986 | 5/21 | 11.61 | 71.93 | 6/09 | 12.81 | 62.10 | 0.00 | 0.00 | 10.73 | 0.09 | 5.85 |
| 1989 | 5/13 | 9.62 | 79.82 | 5/16 | 0 | 68.65 | 0.12 | 0.10 | 0.00 | 0.41 | 0.04 |
| 1997 | 6/06 | 4.62 | 68.27 | 6/10 | 0 | 59.77 | 0.00 | 0.00 | 0.00 | 0.00 | 1.24 |
| 2006 | 5/25 | 6.71 | 72.66 | 6/09 | 61.79 | 72.29 | 0.00 | 0.00 | 0.00 | 0.00 | 2.43 |
| 2008 | 6/09 | 28.72 | 68.54 | 6/10 | 17.25 | 66.53 | 0.00 | 0.00 | 8.71 | 0.07 | 0.81 |
| 2009 | 5/11 | 0 | 77.86 | 5/13 | 0 | 69.47 | 0.10 | 0.05 | 0.00 | 0.41 | 0.05 |



**Table 7 Simulated empty flushing events between 2010 and 2013 based on the optimal strategy with $\alpha = 0.1$**

| Year | Initiation date | Storage in Tsengwen Reservoir at initiation | Storage in Wushanto Reservoir at initiation | Termination date | Storage in Tsengwen Reservoir at termination | Storage in Wushanto Reservoir at termination | Water shortage increment until June | Water shortage ratio until June | Water shortage increment in July | Water shortage ratio in July | Desilting volume |
|------|------|------|------|------|------|------|------|------|------|------|------|
| | | (M. m³) | (M. m³) | | (M. m³) | (M. m³) | (M. m³) | | (M. m³) | | (M. m³) |
| 2010 | 6/04 | 21.02 | 69.74 | 6/08 | 0.00 | 58.70 | 0.00 | 0.00 | 6.37 | 0.18 | 0.74 |
| 2013 | 5/11 | 0.00 | 75.56 | 5/13 | 0.00 | 67.43 | 0.00 | 0.00 | 0.00 | 0.00 | 0.22 |




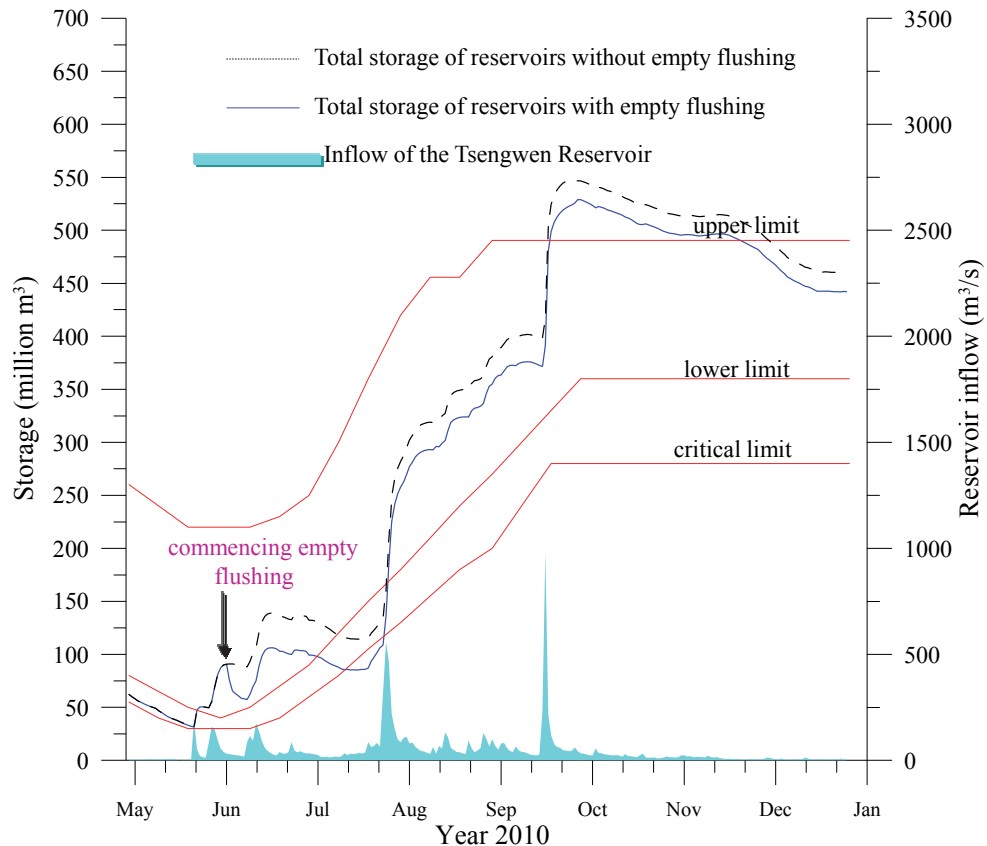

**Fig. 13 Reservoir inflow and storage throughout 2010**





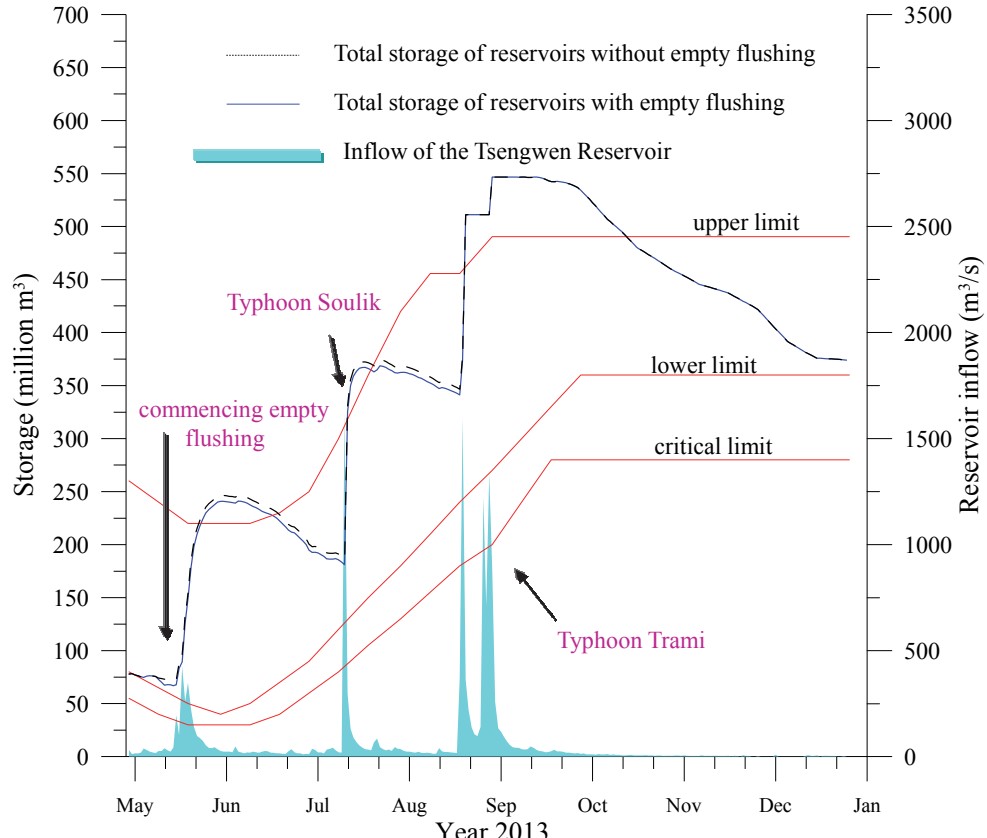

**Fig. 14 Reservoir inflow and storage throughout 2013**

### 4. Conclusions

This study aims to optimize the performance of empty flushing of one primary reservoir within a multi-reservoir system. Prior to empty flushing, the total available storage in a system is allocated from the primary reservoir to the others in order to create favorable initial conditions and prepare backup water supply to be used during empty flushing. The activation and termination conditions of an empty flushing operation are determined according to whether storage in the primary and auxiliary reservoirs satisfy applicable thresholds. Optimization analysis calibrates these storage thresholds to maximize the desilting volume without inducing intolerable water shortage. The case study of the water resources system of



the Tsengwen and Wushanto Reservoirs of southern Taiwan verifies the effectiveness of the derived optimal empty flushing strategy.

For the sake of clarity, the proposed method only discusses systems without means to artificially create inflows for flushing discharge to the primary reservoir. This simplification may not always be valid. For example, there is currently a trans-basin tunnel under construction that will divert surplus water from adjacent basin to the Tsengwen Reservoir in order to enhance the efficiency of regional water utilization. The diverted water could also serve to scour the depositions of Tsengwen Reservoir as well as to replenish the emptied storage following empty flushing. Application of the proposed method to such a system will require optimization of the transferred discharge as a parameter for maximizing the desilting volume during empty flushing. The validity of the employed empirical formula to estimate the discharge of flushed sediments should also be investigated when more field measurements become available. One scenario to which this formula may not apply is drawdown flushing, during which the flow through the outlet is pressurized rather than adhering to open channel flow conditions. Numerical modeling may be employed to more comprehensively and accurately simulate the flushing process in order to optimize the desilting volume.

The operators of Tsengwen Reservoir currently oppose empty flushing due to the high pressure of water shortage, even though reservoir sedimentation imposes a more severe threat in the long term. Nonetheless, this perspective might change with the completion of the sediment sluicing tunnel that is currently under construction as well as the upstream trans-basin diversion tunnel. The design capacity of the sluicing tunnel is 815 $m^3/s$, which enables pre-emptying the reservoir shortly before an expected flood with less uncertainty. It reduces the risk that the inflow is inadequate after the reservoir is emptied. The urgent need of desilting also endows a new role to the conventional projects of water resources development, such as the aforementioned trans-basin diversion tunnel. In addition to elevating the yield of



water supply, it also provides more adequate water to allow recovery and enhanced desilting of existing reservoirs, thus allowing the entire system to advance toward the goal of sustainability.

5    **Acknowledgement**

This work was supported by the Ministry of Science and Technology (Grant No. NSC102-2221-E006-179), Taiwan, R.O.C.

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
