# Peer review of "Manuscript under review for journal Hydrol. Earth Syst. Sci."

_Hydrology and Earth System Sciences, 2016_

## Referee Comment (RC1) · Anonymous Referee #1 · 12 Apr 2016

Review "Assessment of optimal empty flushing strategies in a multi-reservoir system"

General comments: The manuscript describes the selection of an empty flushing strategy applied to a two-reservoir system in Taiwan, first by using a qualitative and quantitative analysis that defines the most suitable period to flush the upper reservoir, and second by optimizing storage thresholds related to the empty-flushing conditions. The main contribution of the paper is on the assessment of empty-flushing in a reservoir system rather than a single reservoir. The paper shows a two reservoir system as a case study which seems insufficient to extend the methodology to multiple-reservoirs systems. The optimization component is also rather small, and it is limited to setting up threshold values after a modified storage-balance curve. In multiple-reservoir systems, this modification might not be that obvious and a complex minimization would be required. The manuscript includes a very useful overview of current cases of empty

flushing experiences in multiple reservoirs, as well as a well-explained list (although it could be summarized for a more easy read) of key factors required to successfully carry out empty flushing. I recommend the publication this manuscript for publication, after minor revisions.

Specific comments: The paper focuses empty-flushing of the primary reservoir, located upstream. Some discussion on how to address the empty-flushing problem for downstream reservoirs should be included. The qualitative analysis does not include any environmental constraints on the problem; neither does the quantitative analysis which probably limits the sediment load discharged. Please explain how the storage balancing curves (example presented in Fig.3) are obtained in the first place; are they based on historic time series without empty-flushing operation? Equation (1) is an important limitation of this study. Some kind of sensitivity analysis would be expected in this case to assess how much it influences the optimized thresholds. This is also somehow captured by the differences shown in Fig.8. Measurements suggest using a $\Psi$ between 2.5 and 10, although for the study used 60 as used by Atkinson. It seems a huge difference with respect to measurements. An important limitation of using the modified storage balancing curve from March until June is losing water at Wushanto. This is not well-explained in the paper. What are the trades-offs of having an increased probability of Tsenweng Reservoir storage dropping below 20M m3? Was the cost function optimized for the period between 1975-2009? This is unclear. For the conclusions, a comment on the use of short-term forecasting and its implementation in Model Predictive Control applications to the case of empty-flushing problems would be appreciated. A more detailed description of the limitations applied in this study, as well as further improvements should be included (the only one mentioned is the empirical formula Eq. 5).

Technical corrections: Page 10, lines 1-5: efficiency of empty flushing ... is between 40 to 60mm? This is not a measurement of efficiency. Fig 1 could be improved by mentioning the parameters to be optimized. Fig 2 could be improved by matching

the notation of X axis with Fig 5 or vice-versa. Also, use a date notation (dd/mmm) to describe vertical lines. How is the water shortage $d_{n,m}$ in Equation 2 computed? Make notation of ($d_i$ Đ̧ $d_j$) in equation 5 clear. What is "Đ̧"? The term Permanent River Outlets (PRO), page 21 line 13, seems not to be often used. Do you mean total outlet structures or the bottom outlet structure? Page 25 line 10: why not until June 30 as it was described in 3.1? Page 26 line 2: identify Rows 3 and 4 with names. Table 2,3,4 could be merged into a single one, taking less space in the paper. Fig. 11: improve readability of lines, which one is which? Legend seems very similar with and without empty flushing.

---

## Referee Comment (RC2) · Anonymous Referee #2 · 9 May 2016

General comments: Reservoir sedimentation management is important worldwide and this paper will contribute to introduce empty flushing operation in multiple reservoirs by considering both flushing efficiency and keeping suitable storage for water supply conditions. This optimization scheme will be valuable in case reservoir sizes are too large to recover storage volume in a short period after emptying flushing.

Specific comments: In the calculation of empty flushing, Eq.(1) is very much important but still unknown parameters are still exist. W $=12.8$ $Q^{0.5}$ and $\psi$ are those key factors. In case higher than EL.185 in Fig.8, we hardly say this is empty flushing and cannot apply Eq.(1) for sediment flushing volume. These are categorized to not free flow flushing but to pressure flushing without fully draw down. If so, there is very limited or almost no data for Tsengwen Reservoir lower than EL.185. This is very much critical defect in this paper. Another issues are environmental constraints. Generally, free flow

flushing should be designed to minimize environmental impacts by high turbidity flow conditions. If so, periodical and short period draw down is suitable. In this regard, every year like in the Kurobe River, Japan or every three years in the Rhone River, Swiss-France are the good examples. Additionally, in order to avoid too much social stress to downstream water users, total duration of empty flushing should be limited less than couples of days because they should stop intake river water during high turbid water passing and they have the maximum acceptable duration for stopping intake. In this paper, if possible, such conditions should be considered. Lastly, one graph which shows sedimentation progress with and without empty flushing should be included.

Technical comments: P3, Line 4, The Tarbela Reservoir is not in Iran but in Pakistan. P21, Line 4, Fig.1 should be changed to Fig.2.

---

## Author Comment (AC1) · 5 Jun 2016

First of all, we wish to express our appreciation for the review and valuable comments from the reviewers. The comments reveal that two major concerns by the reviewers are the lack of validation on the use of the empirical equation to estimate the de-silting volume, and the neglect of the downstream environmental constraints. Our general responses to these concerns are presented in the following two paragraphs respectively. This study attempts to investigate the feasibility of implementing empty flushing to a large reservoir with heavy water supply burden. It is essential that field measurements precisely representing the condition of free surface flushing are currently unavailable, since the reservoir has never gone through such operations. Due to the lack of field data, the flushing coefficient is directly assigned as the most common and conservative value found in literatures. While using a different coefficient

value might lead to a linearly-varied value of the flushed sediment volume (which is the objective function), the optimized storage thresholds should remain unchanged due to the dictated impact by the water shortage constraints. Nonetheless, we do recognize the validity of the employed empirical formula should still be investigated when more field measurements become available, for a more accurate evaluation of the benefits of empty flushing. This issue is emphasized in the first paragraph of subsection 3.6. As for the second concern, the comments of the second reviewer suggest that "in order to minimize environmental impacts by high turbidity flow conditions, periodical and short period draw down is more suitable.... Additionally, in order to avoid too much social stress to downstream water users, total duration of empty flushing should be limited less than couples of days because they should stop intake river water during high turbid water passing and they have the maximum acceptable duration for stopping intake." Although the environmental constraint is not explicitly considered in the case study, fortunately, the patterns of empty flushing from the optimized results conform well to the above required operations. This is because that the strict requirement on the stability of water supply in the case study system has already restricted the frequency and duration of empty flushing. Further, the empty flushing is designed to be performed during the first flood of the wet season. The flood discharge from the downstream watershed is expected to transport the majority of flushed sediments to the downstream receiving water body. Otherwise, the primary reservoir may have to release extra water to assist carrying sediments downstream. In the revised manuscript, the newly added subsection 3.6 serves to more thoroughly present the assumptions and potential future improvements of this study. In this document, we also provide Tables 1 and 2 as point-by-point responses to the reviewers' comments.

Please also note the supplement to this comment:
http://www.hydrol-earth-syst-sci-discuss.net/hess-2016-61/hess-2016-61-AC1-supplement.zip